# Research of High-Purity Lanthanum Prepared by Zone Refining

**DOI:** 10.3390/ma15134603

**Published:** 2022-06-30

**Authors:** Chuang Yu, Bo Pan, Zhiqiang Wang, Dehong Chen, Xiaowei Zhang, Wensheng Yang, Dongwei Zhang, Wenli Lu

**Affiliations:** 1National Engineering Research Center for Rare Earth Materials, GRINM Group Co., Ltd., Beijing 100088, China; yyuchuang@163.com (C.Y.); 13662173988@163.com (B.P.); wzq97122@126.com (Z.W.); chen-dh@126.com (D.C.); 0420295@163.com (X.Z.); yws3179608@163.com (W.Y.); zdw2408268511@163.com (D.Z.); 2GRIREM Advanced Materials Co., Ltd., Beijing 100088, China

**Keywords:** lanthanum, zone refining, SPIM model, the impurity distribution, zone-refining rate, number of passes

## Abstract

In this paper, the purification of lanthanum was studied by using zone-refining technology. The equilibrium distribution coefficient of impurities was calculated using a liquidus slope method to reveal impurities’ distribution properties. Meanwhile, the analysis of impurities’ concentration distribution for Fe and Si has been investigated based on the SPIM model. The calculated findings based on SPIM were found to be in good agreement with the experimental results. In addition, the influence of the zone-refining rate and the number of passes on the purification of lanthanum were studied. It was found that after ten times of zone refining with a zone-refining rate of 5 mm/min, the contents of Fe and Si impurities in metal decreased to 4 and 2 ppm, respectively.

## 1. Introduction

High-purity, rare-earth metals are the material basis for high-performance rare-earth permanent magnets, PV materials, catalysis materials, and their application devices, which have received great interest in both academic and new energy vehicles, integrated circuit, storage chip, 5G, and OLED fields [1]. Therefore, lanthanum has become an increasingly important metal used in modern technology, such as the robot industry and the instrument industry. In the past, it was usually used as an additive in steel, non-ferrous metals, and hydrogen storage alloys, but now it has gradually emerged in emerging industries [2]. The lanthanum-containing BIT(Bi_4_Ti_3_O_12_) thin film is widely used in storage chips because of its unique properties involving high Curie temperature, excellent fatigue resistance, and little leakage current. In recent years, with the continuous development of the storage chip industry, purity requirements have become higher due to the energy efficiency of materials, and it is hard for available products to meet the requirements. Thus, improving the purity of lanthanum to 4N5 (99.995%) is urgent. Compared to other rare-earth elements, lanthanum has a low vapor pressure, which leads to the vacuum distillation method having no effect on its purification. Moreover, the molten salt electrolysis method of reduction reaction caused by current can only improve the purity of lanthanum to 4N (99.99%). The solid-state electromigration method uses a direct current to allow the atoms to move in a certain direction, which is only effective for O, N, etc., but has little effect on metal atoms such as Fe and Al. Among the numerous methods used for purification, such as vacuum distillation, molten salt electrolysis, and solid-state electro-transport methods, the use of zone refining remains the most effective and practical method [3,4].

Many efforts have been made to obtain purer lanthanum. Zong-An Li used solid-state electro-transport to purify La and indicated that metal impurities of Al and Cu in metal La at 800 °C with the direct current are found to significantly migrate to anodes [5]. Wei Wang used La-Ni alloy and La metal as soluble anodes to prepare lanthanum with a purity of 99.867% by molten salt electrolysis [6]. Zhen-Fei Yang found that the purification of metal lanthanum by electron beam smelting can effectively remove metal impurities such as Mg, Li, Mn, Cu, Cr, Fe, Ti, but cannot remove Ni and Si [7]. The zone refining is a powerful method for high-purity metals by using the different solubilities of impurities in the solid and liquid phases, and directional solidification is carried out to remove impurities [8,9]. During the zone-refining process, metals melt and transition into a liquid state under the effect of AC (alternating current), and impurities of k_0_ < 1 (the equilibrium partition coefficient) will move to the end region of the metal, following the zone-refining direction, but impurities of k_0_ > 1 will move in the opposite direction [10]. Heli Wan proposed a modified zone vacuum melting process for producing high-purity aluminum. By modifying the effective redistribution coefficients, axial segregation of impurities was investigated, and the samples with an aluminum content of more than 5 N (99999.2 ppm) were obtained [11]. Zhang Huan developed a novel technique to purify 99.99682% tin by zone refining in an argon atmosphere. The impacts of zone-refining conditions, such as the segregation coefficient, zone-refining rate, and the number of refining passes, on the purity of refined tin were experimentally studied [12]. R. R. Soden prepared nickel single crystals with resistance ratios of 4000 by electron beam float zone refining, and the total carbon and iron content of nickel were less than 5 atom ppm [13]. R. Triboulet used the vertical zone-refining technique to prepare high-purity CdTe [14]. Huang Jun applied induction heating technology to purify industrial cerium and found that the width of the melting zone should be larger in the first few times of zone melting, and a smaller width should be selected in the next few times of zone refining [15]. Nakamura reduced the concentration of the elements with distribution coefficient k > 1 to further improve the purity of ultra-high-purity (99.9999%) aluminum material by the zone-refining method [16]. M Ezheiyan have established a three-dimensional thermal analysis simulation model of a horizontal zone-refining system for germanium semiconductor materials [17]. HK Ji utilized a locally melted zone method by high-frequency induction heating to purify neodymium (Nd) metal and found that a slower traveling rate of the molten zone was more effective for purification [18]. DN Kalievich designed a new construction of zone melting installation to obtain super-pure copper, and optimal parameters of the molten zone width and speed of its movement were defined [19]. Even though the zone-refining technique has been extensively used for the fabrication of high-purity metal, reports on the preparation of lanthanum by zone refining are scarce. The equilibrium partition coefficient of impurities in lanthanum remains largely unknown because of the lack of thermodynamic data.

To obtain high-purity metal lanthanum to meet the development requirements of the storage chip industry, the purification of lanthanum was studied by using zone-refining technology in this paper. Considering the lack of thermodynamic data, the liquidus slope method was utilized to calculate the equilibrium distribution coefficient of impurities for revealing impurities’ distribution properties. Moreover, the migration law of impurities with different characteristics in the process of zone refining was studied by simulating the impurity concentration distribution. The calculated findings based on the SPIM model were found to be in good agreement with the experimental results. In addition, influences of process parameters and conditions on the purification of lanthanum have also been investigated using experimental methods.

## 2. Experimental Procedure

A schematic diagram of the zone-refining device is shown in Figure 1. It can be observed that the whole zone-refining process was completed in a closed furnace. Vacuum pump and argon-purged vials were mounted on the furnace to control the vacuum system. The pressure in the furnace was drawn to 10^−4^ Pa by a vacuum pump and the argon flow rate was maintained at 2 L/min to prevent the oxidation of metals. Samples were placed in a water-cooled crucible and heated by a coil. During the experiment, water will continuously flow through the crucible to cool it down. The power was generally set at 30 KHz and the temperature was set to 1050 °C. After the power setting, the AC moved through the coil and created an electromagnetic field, which caused melting of metal. The drive motor was used as a transmission to control the movement of the crucible to vary the heating location.

The 4N raw materials used in this study were prepared by LeShan Grinm Advanced materials using the molten salt electrolysis technique, which was processed into a round bar with a length of L = 20 cm and a radius of 3 cm and placed in the crucible. Four samples of lanthanum were analyzed individually by glow discharge mass spectrometry (GDMS) and its specific components are shown in Table 1. It can be seen from Table 1 that Fe and Si impurities have high concentrations, which have a large effect on the purity of the metal, and more attention needs to be paid to them during the whole procedure. In contrast, other impurities in lanthanum samples are lower or negligible. The analysis results show that the impurity concentration difference between the four samples is small, which indicates that the uniformity of electrolytic lanthanum is good. In cases where the difference was smaller, the mean of four results was accepted as the initial concentration. Furthermore, the bulk lanthanum was then melted and poured into the mold of the same size as the crucible. After cooling, lanthanum was moved to the crucible and zone-refining experiments were carried out as previously described.

## 3. Results and Discussion

### 3.1. Calculation of Equilibrium Distribution Coefficient

The equilibrium distribution coefficient (*k*_0_) is defined as *k*_0_
*= C_S_/C_L_*, where *C_S_* and *C_L_* represent the solubilities of impurities in the solid phase and liquid phase, respectively. During the zone-refining process, impurities of *k*_0_ < 1 will move to the end of the metal, following the zone-refining direction, but impurities of *k*_0_ > 1 will move in the opposite direction. When the value of *k*_0_ is very close to 1, the effect of purification is weakened and no longer works. Conversely, impurities’ concentrations after zone refining are lower the further *k*_0_ is from 1. As the purity of lanthanum used in the experiment is relatively high, interaction energies between impurities are weak. It suggests that the equilibrium partition coefficient of impurities can be replaced by the corresponding equilibrium partition coefficient of the binary system. By measuring the liquidus and solidus slopes of the binary phase diagram, the corresponding equilibrium distribution coefficient is calculated. However, the solidus slopes are extremely difficult to accurately measure, which makes it difficult to obtain an accurate value of *k*_0_ with this method. Based on the reasons listed above, liquidus slopes ($m_{L}$) combined with latent heat of crystallization ($\Delta H_{m}$) were used to calculate the corresponding equilibrium distribution coefficient of impurities in this paper [20]:(1)$$k_{0}=1+\frac{\Delta H_{m}}{RT_{m}^{2}}m_{L}$$
where *R* is the molar gas constant, $\Delta H_{m}$ is the latent heat of crystallization of lanthanum, and $T_{m}$ represents the melting point of lanthanum. The relevant thermodynamic parameters are listed in Table 2.

By using the equilib module of Factsage software (Montreal, Canada), the binary phase diagrams of La-Fe and La-Si were obtained, as shown in Figure 2 and Figure 3.

*k*_0_ is an important basis for judging the migration direction of impurities, which would be influenced by other factors during the zone-refining process. Thus, the effective partition coefficient (*k*_eff_) is introduced to more accurately reflect the migration characteristics of impurities. The relationship between *k*_0_ and *k*_eff_ was expressed as follows [21]:(2)$$k_{\mathrm{eff}}=\frac{k_{0}}{k_{0}+\left(1-k_{0}\right)e^{\left(-v\delta /D\right)}}$$

In Equation (2), *v* is the zone-refining rate, *D* is the diffusion coefficient of impurity, and δ is the diffusion layer thickness. Normally, δ = 1 mm in the existence of convection, and it can be reduced to 0.1 mm if mixing exists. Since the zone-refining experiment is completed by the high-frequency current, the induced magnetic field will produce a Lorentz force on the molten zone and play a role of mechanical stirring, and δ can be considered as 0.1 mm. The relationship between the zone-refining rate and effective partition coefficients can be seen in Table 3.

As can be observed from Table 3, the effective partition coefficients of Fe impurities were 0.150133, 0.384395, and 0.688194, respectively, while the effective partition coefficients of Si impurities were 0.119956, 0.164083, and 0.220380, respectively, when the zone melting rate was 5, 15, and 25 mm/min. However, the effective partition coefficients of impurities increased gradually with the increase of the zone-refining rate. In general, this phenomenon is unfavorable for efficient removal of impurities. Compared with larger rates, the effective partition coefficients were closer to the equilibrium distribution coefficient when the rate was small. By calculating the effective partition coefficients of impurities, it was revealed that *k*_eff_ values of Fe and Si were less than 1 under different conditions, which confirmed that the removal of Fe and Si impurities in lanthanum by zone melting is both feasible and effective.

### 3.2. The Effect of Zone-Refining Rate on Impurity Distribution

Zone refining can be considered as the crystallization process. Through the recrystallization process after melting, the distribution of impurity concentration regularly changes with the position. Further, the Lorentz force plays a major role in the stirring of the molten zone. The impurity redistribution in the liquid is powerfully supported by convection. In contrast, the diffusion plays a minor role during the process of impurity migration. To better investigate the distribution of impurities, it was necessary to make the following assumptions:The parameter *k*_eff_ is assumed to be constant during zone refining.The widths of the molten zone, cross-sectional area, zone-refining rate, temperature, and other relevant process parameters remained unaltered and stable.Impurities are uniformly distributed over the whole molten zone area without segregation.The diffusion of impurities in the solid phase is negligible.The solidification interface is flat and in equilibrium.

The theoretical model of impurity distribution is evolved from the Scheil equation, and the SPIM model studies the relationship between the impurity concentration of solid phase $C_{S}$ and the initial concentration $C_{0}$ with the position after a single zone refining based on mass-conservation principles:(3)$$\frac{C_{S}}{C_{0}}=1-(1-k_{\mathrm{eff}})\exp\left(-\frac{k_{\mathrm{eff}}X}{Z}\right),\;0<X<1-L$$
(4)$$\begin{array}{c} \begin{array}{cc} \frac{C_{S}}{C_{0}}= & \left\{1-(1-k_{\mathrm{eff}})\exp\left[-\frac{k_{\mathrm{eff}}(1-Z)}{Z}\right]\right\}\times {\left\{1-\left[\frac{X-(1-Z)}{Z}\right]\right\}}^{k_{\mathrm{eff}}-1} \end{array},\\ \;1-L\leq X\leq 1-L \end{array}$$
as shown in the Figure 4, where L represents the length of the whole sample, which is 20 cm, *Z* is the width of the molten zone, *Z* = 2 cm, and *x* is the distance that the molten zone has moved. Here, the data were normalized, L is considered to be 1, $Ζ=\frac{z}{L}<1$, and $X=\frac{x}{L}\leq 1$.

The multiple zone-refining equation is different from the single zone melting equation. The whole sample can be divided into four regions for analysis. Region 1: surface (*X* = 0), region 2: intermediate (0 < *X* ≤ 1 − *Z*), region 3: normal freezing (1 − *Z* < *X* < 1), and region 4: end of the sample (*X* = 1), and the corresponding equations for each region are described below:(5)$$C_{S(0)}^{n}=k_{i}\left(\frac{dx}{Z}\right)\left(\overset{M-1}{\underset{q=0}{\sum }}C_{S(qdx)}^{n-1}\right),\;X=0$$
(6)$$C_{S(X)}^{n}=C_{S(X-\mathrm{dx})}^{n}+\frac{k_{i}\;dx}{Z}\left[C_{S(X+Z-dx)}^{n-1}-C_{S(X-dx)}^{n}\right],\;0\;<\;X\;\leq \;1-Z$$
(7)$$C_{L}^{n}=\frac{C_{0}}{Z}-\frac{dx}{Z}\left(\overset{M-1}{\underset{q=0}{\sum }}C_{S_{\left(q\mathrm{dx}\right)}}^{n}\right),\;1-Z\;<\;X\;<\;1$$
(8)$$C_{S_{1}}^{n}=C_{0}-\frac{\sum_{q=0}^{M-1}C_{S_{\left(qdx\right)}}^{n}}{M},\;X=1$$
where *N* represents the number of zone-refining passes, and *M* is the number of single elements. A more detailed derivation can be found in [22].

The zone melting experiment was carried out using the device shown in Figure 1 and the zone melting rates were set as 5, 15, and 25 mm/min, respectively. After a zone refining, five different sample positions of the La were taken for impurity content analysis, as shown in Figure 5. The impurity distribution under the same experimental parameters was calculated by using the SPIM model and compared with the experimental results.

It can be observed from Figure 6 and Figure 7 that the concentrations of impurities in the metal gradually increased from the beginning to the end after zone refining and increased drastically in the last region (1 – Z < *X* < 1). This is due to the effective partition coefficients of impurities Fe and Si in the zone melting process being less than 1, which tended to exist in the liquid phase. With the process of zone refining, impurities would gradually migrate from the solid phase into the liquid phase and moved to the tail of the sample with the molten zone, causing tail enrichment. In addition, the concentration of impurities in the front region (0 < *x* < 0.5) decreased with the decrease of the zone-refining rate, which indicates that the purification effect has been enhanced. When the moving rate of the molten zone is too fast, impurities do not have enough time for effective migration, which weakens the purification efficiency. Meanwhile, it was found that when the zone-refining rate increased from 5 to 25 mm/min, the Fe impurity concentration at *x* = 0.1 increased from 15 to 46 ppm, while the Si impurity concentration at the same position only increased from 11 to 14 ppm. The influence of the zone-refining rate on the variation of Fe was much greater than that of Si. This is because the diffusion rate of Fe is much lower than that of Si, which requires more time for migration in the same condition. Therefore, when the zone melting rate increases, the time left for impurity migration is shortened accordingly. With the increase of the zone-refining rate from 5 to 25 mm/min, the effective partition coefficient of Fe increased from 0.150133 to 0.688194, while Si only changed from 0.119956 to 0.220380. The closer to 1 the equilibrium distribution coefficient is, the more difficult the removal of impurities. Thus, we can conclude that Si is easier to remove than Fe in lanthanum.

### 3.3. The Effect of the Number of Refining Passes on Impurity Distribution

To further improve the purity of lanthanum, the distribution of impurities under multi-pass zone refining when the zone-refining rate was 5 mm/min has been investigated. As shown in Figure 8 and Figure 9, the impurity concentration of the region 0 < *x* < 0.7 decreased gradually with the increase of zone-refining passes with the ongoing zone-refining process. However, it did not work in the region 0.7 < *x* < 1. After three zone melting experiments, the impurity Fe concentration at *x* = 0.1 was only 4.2 ppm, while Si was also reduced to 1.7 ppm, which was significantly reduced compared with the content of raw materials. Whereas at *X* = 0.9, the Fe impurity concentrations were 84, 96, and 90 ppm, respectively, while Si concentrations were 74, 84, and 88 ppm, respectively, after three times of zone melting. The results between the three experiments differed by a modest amount. During zone refining, impurities will migrate from top to tail because of the effect of segregation, and diffuse from high concentration to low concentration due to concentration differences at the same time. After long-term theoretical practice, it has been found that impurities cannot be completely removed, and the concentration will reach a relatively stable distribution after multiple zone refining. Under the influence of segregation together with diffusion, impurities are in dynamic equilibrium and the concentration approaches the limiting distribution.

The calculated values were in good agreement with the experiment in region 0 < *X* < 0.7, but were much lower than the experiment values in *X* = 0.9. It can be found that the impurity concentration after three times of zone refining was higher than that after one time of zone refining in *X* = 0.9. After multi-pass zone-refining experiments, impurities were enriched at the tail of the metal and much higher than the top. At this time, the segregation effect was weakened, but the diffusion effect caused by the concentration difference played a dominant role. Some tail impurities would be re-entered into the molten region through the melting interface and spread throughout the whole region through the molten region mixing, which is called backflow. As a result, the impurity concentration increases, and the purification effect is not obvious.

Due to the enrichment of impurities in the tail, the high-purity lanthanum was obtained after cutting off the tail part in the practical application. Here, the average concentrations of *x* = 0.1, 0.3, and 0.5 were chosen to represent the residual concentration of impurities after zone refining. Figure 10 shows the residual concentration of Fe and Si impurities when the number of zone melting was 5, 10, and 15 times, respectively. As can be observed from Figure 10, the residual impurity Fe content in the metal was 6 ppm, while Si was 2.9 ppm when the number of refining passes was 5. The Fe content was reduced to 4 ppm, and the Si content was reduced to 2 ppm when the number of refining passes was 10. However, when the number of refining passes continued to increase to 15, Fe was only reduced to 4.2 ppm and Si was 1.8 ppm, which were not significantly different from 10 passes. At this time, the impurities have approached the limiting distribution, and increasing the number of refining passes did not show a large impact on the purification effect.

## 4. Conclusions

In this paper, the purification process of lanthanum has been analyzed by using zone-refining technology. The equilibrium distribution coefficient of impurities was calculated using a liquidus slope method to reveal impurities’ distribution properties. In addition, influences of the zone-refining rate and the number of passes on the purification of lanthanum were investigated. The conclusions were as follows:
By calculating the effective partition coefficients of impurities, it was revealed that *k*_eff_ of Fe and Si was less than 1 under different conditions, which confirmed that the removal of Fe and Si impurities in lanthanum by zone refining is both feasible and effective.The concentrations of Fe and Si in the metal gradually increased from the beginning to the end after zone refining and increased drastically in the last region. The purification effect will increase with the decrease of the zone-refining rate, and Si was easier to remove than Fe in lanthanum.With the increase of the number of zone refining, the purification effect increased significantly, but impurities approached the limiting distribution at 10 passes. Increasing the number of refining passes did not show a large impact on the purification effect.

## Figures and Tables

**Figure 1 materials-15-04603-f001:**
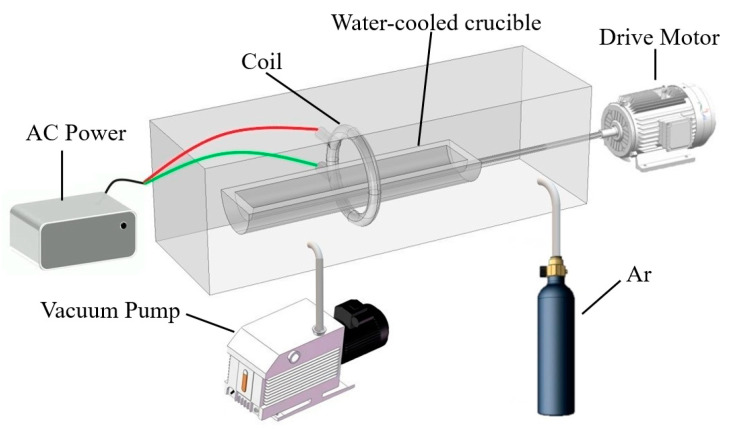
The schematic diagram of the zone-refining device.

**Figure 2 materials-15-04603-f002:**
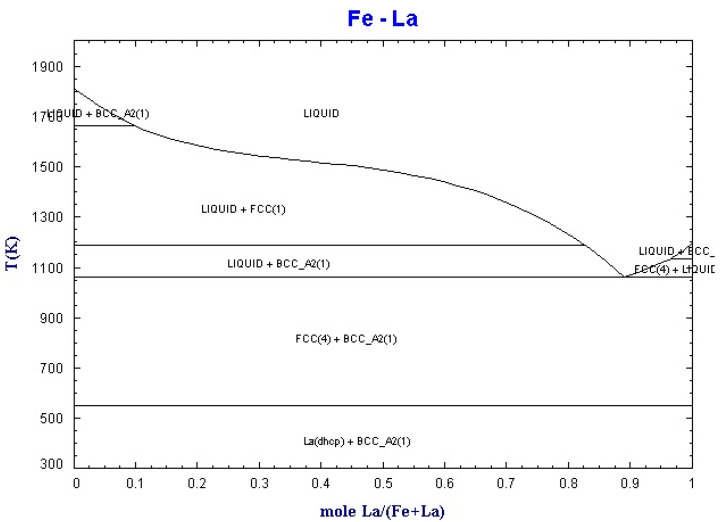
Binary phase diagram of La-Fe.

**Figure 3 materials-15-04603-f003:**
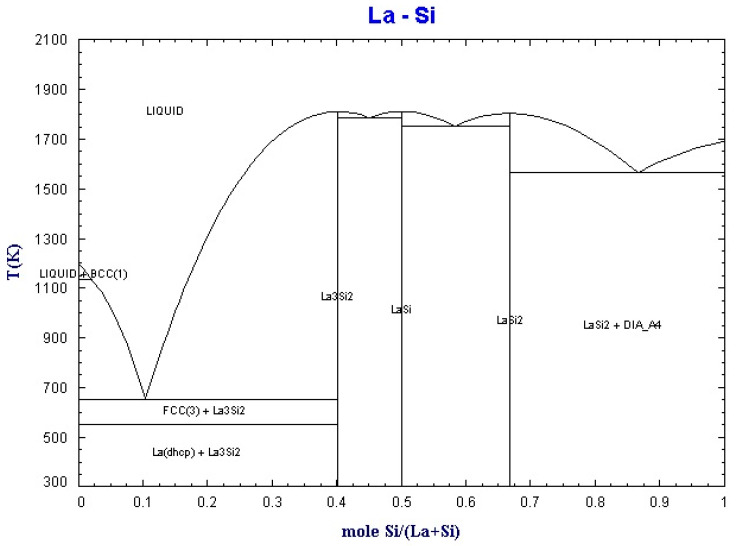
Binary phase diagram of La-Si.

**Figure 4 materials-15-04603-f004:**
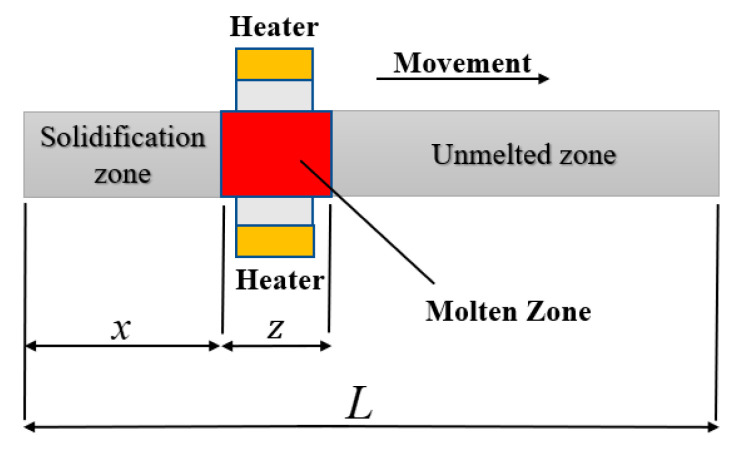
The schematic diagram of zone refining.

**Figure 5 materials-15-04603-f005:**
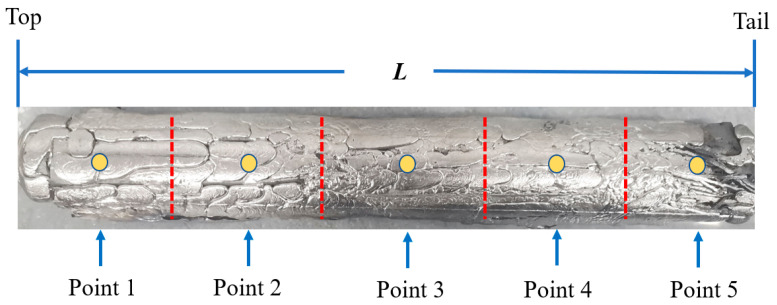
The schematic diagram of analysis location after zone refining.

**Figure 6 materials-15-04603-f006:**
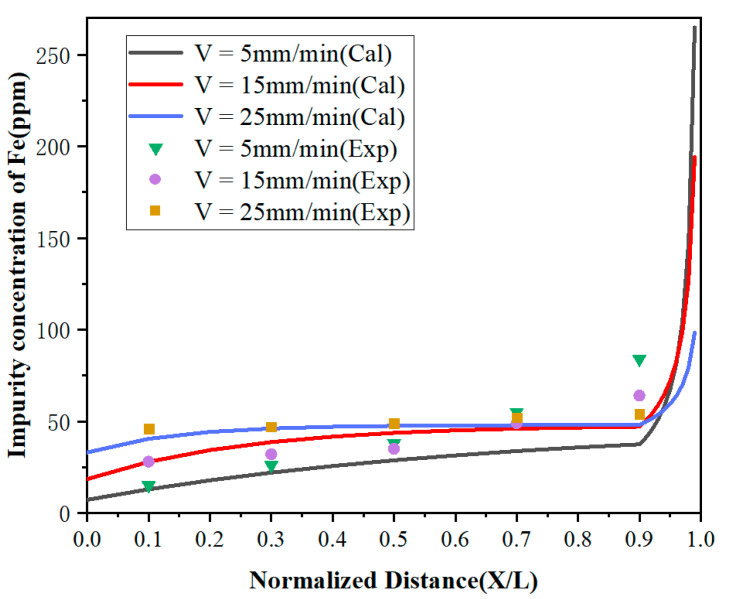
Distribution of Fe under different zone-refining rates.

**Figure 7 materials-15-04603-f007:**
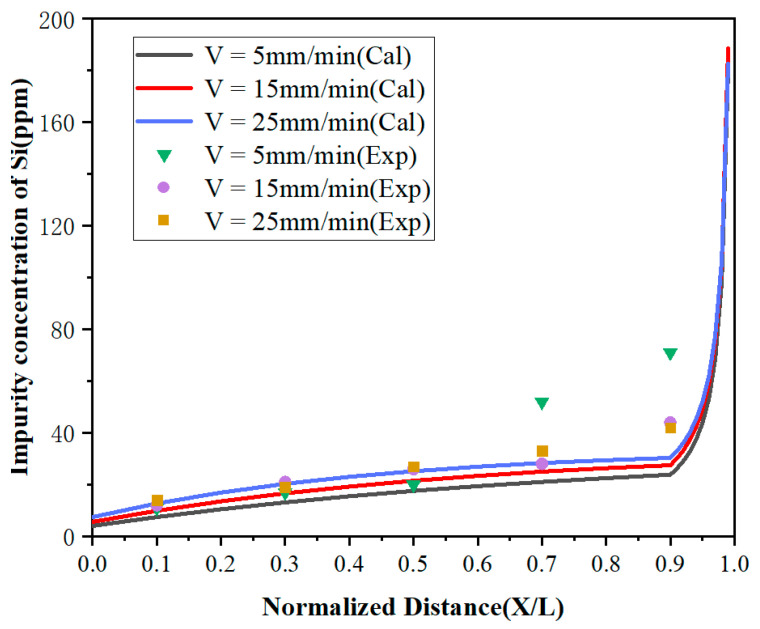
Distribution of Si under different zone-refining rates.

**Figure 8 materials-15-04603-f008:**
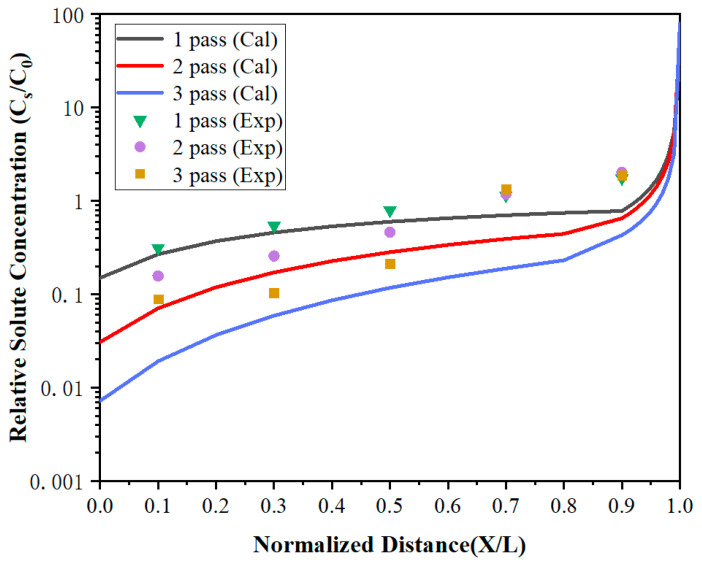
The distribution of Fe impurities under multi-pass zone refining.

**Figure 9 materials-15-04603-f009:**
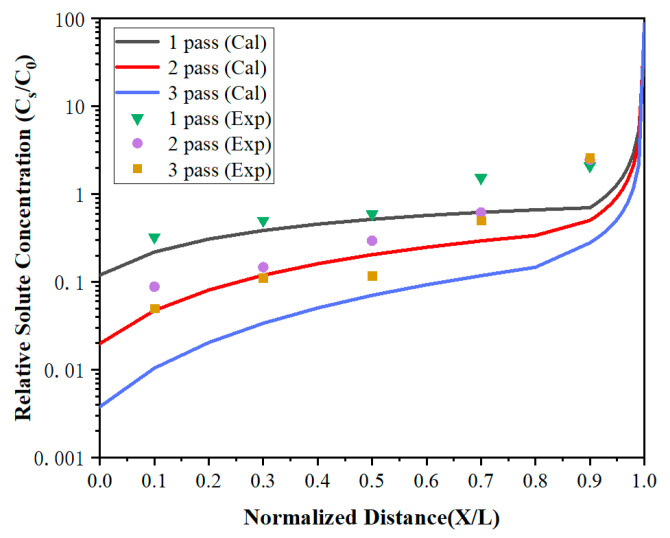
The distribution of Si impurities under multi-pass zone refining.

**Figure 10 materials-15-04603-f010:**
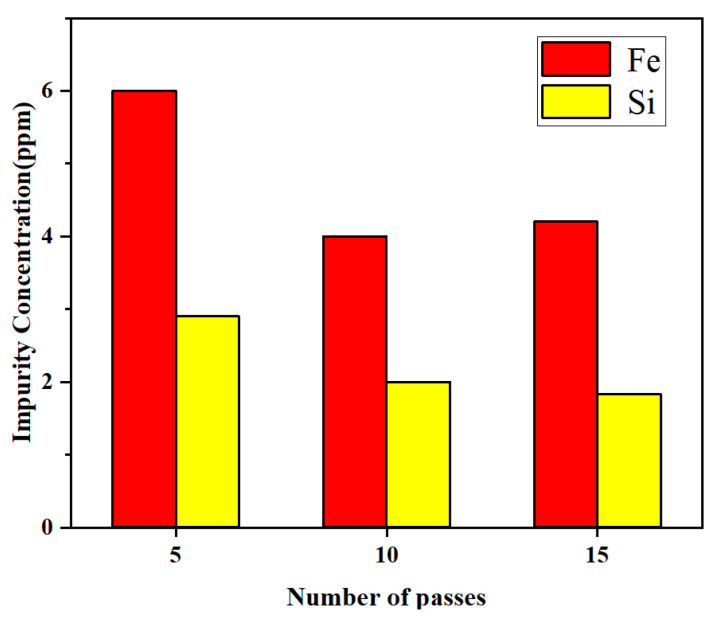
Relationship between the residual concentration of impurities and the number of refining passes.

**Table 1 materials-15-04603-t001:** GDMS analysis of impurity concentrations (ppm).

Impurity	1	2	3	4	Average
Fe	46	47	49	50	48
Cu	0.3	0.5	0.6	0.6	0.5
Cr	0.65	0.32	0.51	0.6	0.52
Si	39	31	25	41	34
Mn	0.015	0.014	0.013	0.018	0.015
Zn	0.003	0.007	0.006	0.008	0.006
Ti	0.004	0.001	0.005	0.006	0.004
Pr	0.003	0.001	0.002	0.006	0.003
Ce	0.11	0.13	0.15	0.09	0.12
Co	0.012	0.007	0.004	0.005	0.007

**Table 2 materials-15-04603-t002:** The equilibrium distribution coefficients of impurities.

Metal	$\Delta H_{m}$	$T_{m}$	Impurity	$m_{L}$	*k* _0_
La	6196.50 J/mol	1193.15 K	Fe	−1746.03	0.0859
			Si	−1715.26	0.102

**Table 3 materials-15-04603-t003:** Effective partition coefficients of impurities.

Impurity	*D*	*v*	$k_{\mathrm{eff}}$
Fe	1.32 × 10^−8^m^2^s^−1^	5 mm/min	0.150133
		15 mm/min	0.384395
		25 mm/min	0.688194
Si	3.47 × 10^−8^m^2^s^−1^	5 mm/min	0.119956
		15 mm/min	0.164083
		25 mm/min	0.220380

## Data Availability

Data sharing is not applicable to this article.

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
