# Peer review of "Research of High-Purity Lanthanum Prepared by Zone Refining"

_materials, 2022, doi:10.3390/ma15134603_

Round 1

Reviewer 1 Report

The article entitled "Research of High-Purity Lanthanum Prepared by Zone Refining" is quite interesting since it studies the improvement of the purity of lanthanum. However, the article requires changes before being accepted for publication, particularly regarding English language. Some comments are listed below:

Line 15: add "been"  after "have"

Line 28: "metal"

Line 30: please mention some of the emerging industries.

Please define the abbreviations the first time that they appear in the text

Line 33: what requirements?

Line 35: what purity?

Line 37: please indicate some other competing methods:

Introduction in general: some regining methods applied in the field of rare earths should be indicated as well as details of the mechanisms

Is there any research about refining lanthanum in any manner? I assume that it is possible to find some, or at least some methods of refining rare earth metals. With a simple search in google it is possible to find some research in the field of rare earth elements. Please provide further references and details about the refining in this field.

Further information should be given about the manufacture of lanthanum process, also about the purity obtained at the end of these processes (not only a simple sentence as line 84, something more detailed)

Line 75: "was" after "which". This comments are general but English should be carefully reviewed.

Line 106: How this can be justified?

keff is the effective partition coefficient, please mention it the first time you write it

Pay attention to the format in general, particularly subscripts or ","

Further justification on the assumptions made in section 3.2 should be given

Line 195: What is too large? Please clarify

Lines 220-222: These differences are real or within the error margin?

Reviewer 2 Report

The paper is well written. The author analyses the results well in the present format.

The references are not adequate. There are plenty of papers available in zone refining which authors have missed. I request the authors to add the valid scientific paper with diversity and not from the same country of origin.

 page 1, line 43-45 can be removed or re phrase

With respect to zone refining the authors have discussed only very little references. Kindly include more references in the Introduction sections

page 2, line 62 -70 is the repeat of abstract. Kindly modify

Fig 1, setup furnace details are not mentioned  properly. Kindly include the details.

page 2, line 77-80 can be removed

Sample used in the study details are missing. need more discussion in the directions.

Sample size and temperature details not yet mentioned clearly.

page 3. lines 101-104 justification?

The eqn and the values need to be double check. On what basis the author choose the table 2 and 3?

Figure 5, on what basis the authors choose the line and points. What is the total L?

There are plenty of graphs and the authors have explained the discussion not appropriately. Most of the results were reported not discussed scientifically.

Round 2

Reviewer 2 Report

In the revised article, the author made the expected changes.
The article is recommended to consider for publication to Editor.